# Recent Progress in Oculopharyngeal Muscular Dystrophy

**DOI:** 10.3390/jcm10071375

**Published:** 2021-03-29

**Authors:** Satoshi Yamashita

**Affiliations:** Department of Neurology, Graduate School of Medical Sciences, Kumamoto University, Kumamoto 860-8556, Japan; y-stsh@kumamoto-u.ac.jp; Tel.: +81-96-373-5893

**Keywords:** oculopharyngeal muscular dystrophy, clinical characteristics, pathogenesis, therapeutic approach, patient registry

## Abstract

Oculopharyngeal muscular dystrophy (OPMD) is a late-onset intractable myopathy, characterized by slowly progressive ptosis, dysphagia, and proximal limb weakness. It is caused by the abnormal expansion of the alanine-encoding (GCN)n trinucleotide repeat in the exon 1 of the *polyadenosine (poly[A]) binding protein nuclear 1* gene (11–18 repeats in OPMD instead of the normal 10 repeats). As the disease progresses, the patients gradually develop a feeling of suffocation, regurgitation of food, and aspiration pneumonia, although the initial symptoms and the progression patterns vary among the patients. Autologous myoblast transplantation may provide therapeutic benefits by reducing swallowing problems in these patients. Therefore, it is important to assemble information on such patients for the introduction of effective treatments in nonendemic areas. Herein, we present a concise review of recent progress in clinical and pathological studies of OPMD and introduce an idea for setting up a nation-wide OPMD disease registry in Japan. Since it is important to understand patients’ unmet medical needs, realize therapeutically targetable symptoms, and identify indices of therapeutic efficacy, our attempt to establish a unique patient registry of OPMD will be a helpful tool to address these urgent issues.

## 1. Introduction

Oculopharyngeal muscular dystrophy (OPMD), a late-onset myopathy, is characterized by slowly progressive ptosis, dysphagia, and proximal limb weakness. The disease is caused by the abnormal expansion of (GCN)n repeats (11–18 repeats in the disease condition vs 10 repeats in normal condition) in exon 1 of the *poly (A) binding protein, nuclear 1 (PABPN1)* gene [1]. OPMD is a rare form of muscular dystrophy, whose prevalence remains unknown in the non-endemic areas including Japan. According to the details available on the website ClinicalTrials.gov [2], 12 clinical studies on OPMD are ongoing at the moment. The outcome of a few of these studies may provide therapeutic benefits in alleviating the swallowing problems in these patients. Thus, there is an urgent need to assemble information regarding OPMD patients for the introduction of such effective treatment strategies. In this article, we present recent advances in OPMD research and introduce an idea for the setup of a nation-wide OPMD disease registry in Japan.

## 2. Epidemiology

OPMD is distributed worldwide, however, the prevalence of the disease varies in different ethnicities. The highest prevalence has been reported in the Bukhara Jews in Israel (1:600) [3], French-Canadians (1:1000) [4], and Hispanic in New Mexico [5], whereas the prevalence is 1:100,000–1,000,000 in the European population [6,7]. So far, case reports of a relatively small number of patients have been published from Italy [8], France [9], Germany [10], Uruguay [11], England [12], the Netherlands [13], Scotland [14], Spain [15], Denmark [16], Bulgaria [17], mainland China [18,19], Taiwan [20], Hong Kong [21], South Korea [22], Thailand [23], Malaysia [24], Singapore [25], Japan [26,27,28], and others. However, the precise epidemiological and clinicopathological characteristics of OPMD have not yet been investigated in Japan. In our single-institution experience, only six patients with OPMD were diagnosed during the last decade for which 414 patients underwent muscle biopsy.

## 3. Presentation

Major symptoms of OPMD are ptosis, swallowing dysfunction, and proximal limb weakness, although the frequency of appearance of these symptoms varies in different ethnicities. Previous reports revealed that almost all OPMD patients showed ptosis, whereas 62–100% and 20–81% of the patients showed swallowing dysfunction and proximal limb weakness, respectively [13]. The initial symptom is ptosis in approximately two-thirds of the OPMD patients, accompanied by ocular motor abnormalities in half of the patients [12]. Subsequent symptoms include lower proximal limb weakness followed by dysphagia. Since some patients develop lower proximal limb weakness alone [13], it may lead to a misdiagnosis of limb-girdle muscular dystrophy or myositis. A recent study revealed the possibility that the length of (GCN)n repeats defines the disease severity and progression [29]; patients with the longer (GCN)n repeats were diagnosed at an early age.

Patients with OPMD can manifest extramuscular symptoms (Figure 1), such as deterioration of respiratory function, including decreased forced expiratory volume in 1 s [30], although cardiac functions may be preserved. Coexistence with dementia [31,32], executive function deficits [33], or peripheral neuropathy [34] was reported. Approximately half of the patients complained of fatigue and pain, and most of the patients were impaired in daily life activities, social participation, and ambulation [35].

The autosomal recessive form of OPMD, which is caused by the homozygous expansion of (GCN)11 repeats in the *PABPN1* gene, has been reported as late-onset and less severe than the dominant form [1], although it can manifest various symptoms with an unusual onset and atypical clinical course [36].

## 4. Diagnosis

### 4.1. Blood Examination

Serum levels of creatine kinase (CK) are slightly elevated in mild cases, but the levels are highly increased in more severe cases [29]. However, the levels of myogenic enzymes gradually decrease and are normalized as the disease advances (unpublished personal data), suggesting the limitation of using the enzymes as biomarkers for evaluating disease progression and therapeutic efficacy.

### 4.2. Electrophysiology

Needle electromyography demonstrated low-amplitude and short-duration motor unit potentials in the affected muscles in almost all the cases, although the electromyographic changes are non-specific for the disease [37]. Some case series reported the coexistence of peripheral neuropathy; other reports showed a low proportion of cases with a low amplitude of sensory nerve action potential on the sural nerves, while most of the cases had no abnormalities [37,38].

### 4.3. Histology

The muscle histological features are characterized by varied fiber size, small rounded or angular fibers, and rimmed vacuoles (RV) in small fibers, although the frequency of fibers with RVs is only 0.6% and not necessarily all the patients show these changes [39]. Necrotic and regenerative myofibers are rarely observed. Mitochondrial abnormality has been reported, as evidenced by COX-negative fibers and ragged-red fibers [40,41,42]. A more specific finding is intranuclear tubulofilamentous aggregates of 8.5 nm outer diameter, which were observed approximately in 4% of the myonuclei by electron microscopy [39]. Of note, the detection of PABPN1-positive insoluble intranuclear aggregates was reportedly 100% sensitive and 96% specific for OPMD diagnosis [43].

### 4.4. Radiology

Recent evidence regarding radiological findings of OPMD has been accumulating. In previous studies with a relatively small number of patients, radiological features in magnetic resonance imaging (MRI) of OPMD patients included fatty replacement prominent in the adductor of the thigh, hamstring, soleus, gastrocnemius muscles, and muscles regulating the swallowing process including the tongue muscles [44,45]. A recent imaging study on 168 patients identified fatty replacement in 96.7% of all symptomatic patients, particularly in the most commonly affected muscles, including the tongue, adductor magnus, and soleus muscles [46]. In the same study, muscle pathology on MRI reportedly correlated with disease duration and functional impairment. Another study observed a highly negative correlation between clinical scores and visual imaging scores [44].

### 4.5. Swallowing Examination

Several studies have been conducted to figure out the mechanisms of dysphagia in OPMD. A manometric study showed abnormalities with simultaneous contractions and incomplete lower esophageal relaxation, as well as impaired upper esophageal sphincter/pharyngeal pressure [47]. A preliminary videofluorographic study revealed reduced pharyngeal constriction, incomplete laryngeal vestibule closure, and decreased speed and range of hyoid movement [48]. A more recent study showed that the proportions of the patients with pharyngeal dysphagia, cricopharyngeal bar, vallecular residue, and piriform sinus residue were 96%, 45%, 77.3%, and 90.1%, respectively [49]. In our investigation, the fiber-optic endoscopic evaluation identified a significant impairment in pharyngeal clearance resulting in salivary pooling (data submitted). Our videofluorographic evaluation showed severe disturbances in tongue base retraction and pharyngeal constriction and clearance.

### 4.6. Genetic Examination

The underlying cause of OPMD is the abnormal expansion of (GCN)n repeats (11–18 repeats in the disease condition, while 10 repeats in normal condition) in the *PABPN1* gene, located on the chromosome 14 [1]. One of the parents may have the mutation in the autosomal dominant form or both parents may have the (GCN)11 repeats expansion in the autosomal recessive form.

### 4.7. Diagnosis

OPMD diagnosis is based on typical clinical symptoms and molecular genetic testing. Patients with slowly progressive ptosis and dysphagia symptoms at onset after the age of 40 years, and positive family history, should be considered applicable for DNA sequencing, which is a gold standard for OPMD diagnosis [50]. Histological and electromyographic studies can be performed to exclude the possibility of other neuromuscular disorders.

## 5. Pathogenesis and Therapeutic Approach

Wild-type PABPN1 protein shuttles between the nuclei and the cytosol and play important roles in polyadenylation through enhancing the processivity of the poly(A) polymerase, modulating alternative polyadenylation sites, aiding nuclear RNA export, and regulating the steady-state level of long non-coding RNAs [51].

In patients with OPMD, the PABPN1 protein has an extra 11–18 alanine residues in the N-terminus due to the expanded (GCN)n repeats in the exon 1 of the *PABPN1* gene. The following underlying mechanisms of OPMD have been speculated: (1) the mutated protein forms intranuclear aggregates, leading to toxicity; (2) the intranuclear aggregates sequester various transcription factors, molecular chaperones, RNA binding proteins, and RNAs necessary for cell maintenance; (3) reduction in the levels of wild-type PABPN1 by half suppresses its native function; or (4) the abnormal PABPN1 protein suppresses the function of wild-type protein [51]. Recent progress in therapeutic strategies for the treatment of OPMD is described below (Figure 2).

### 5.1. Knockdown of Mutant PABPN1 and/or Augmentation of Wild-Type PABPN1

In a mice model of OPMD, co-delivery of two adeno-associated virus (AAV) vectors, the first expressing three shRNAs under the control of RNA polymerase III promoter and the second expressing wild-type human PABPN1 under the control of a skeletal and cardiac muscle-specific promoter significantly improved the histopathological features [52]. Using a dual AAV vector expressing both the cassettes also produced similar effects, suggesting a possible clinical application of gene therapy [53].

### 5.2. Boosting of Muscle Growth

Systemic delivery of a monoclonal antibody that inhibits myostatin in a mouse model of OPMD improved the body and muscle mass, increased the muscle strength and the myofiber diameter, and reduced the expression of fibrosis markers, although no effect on intranuclear inclusions was observed [54]. However, a more recent study showed that inhibition of myostatin failed to revert the muscle atrophy in OPMD mouse model but effectively reduced the expression of histological markers of fibrosis in the treated muscles [55]. Based on these studies, it could be concluded that inhibition of myostatin commonly prevents muscle fibrosis.

### 5.3. Reduction and Dispersal of Intranuclear Aggregates

Since the intranuclear aggregates of the mutant PABPN1 protein might be toxic, pharmacological approaches to specifically target these aggregates using cystamine [56], doxycycline [57], trehalose [58], guanabenz [59], and intrabodies [60] successfully ameliorated the disease in several models of OPMD. The soluble form of the mutant PABPN1 might also produce toxicity [61] by interacting with the components of the transcription complex/histone acetylation [62]. Valproic acid, a direct inhibitor of histone deacetylase classes I and II, could ameliorate the mutant PABPN1 toxicity in cellular and worm models of OPMD by enhancing the level of histone acetylation [63]. Further studies are required to understand the disease pathogenesis involving the soluble form of mutant PABPN1.

### 5.4. Autologous Myoblast Transplantation

A previous study showed that myoblasts isolated from the unaffected muscles of OPMD patients proliferated and differentiated normally, whereas myoblasts isolated from the affected muscles (cricopharyngeal muscle) exhibited reduced myogenicity [64]. A phase I/IIa clinical study based on the grafting of autologous myoblasts isolated from unaffected muscles (quadriceps or sternocleidomastoid) into the pharyngeal muscles following a cricopharyngeal myotomy showed short and long-term safety, tolerability, an improvement in the quality of life score, and no functional degradation in swallowing [65]. As of now, autologous myoblast transplantation stands on the verge of clinical application.

### 5.5. Mitochondrial Restoration

Mitochondrial abnormality has been proposed as the possible etiology, although the molecular pathogenesis remains unclear. A recent knock-in mouse model containing an alanine-expanded *PABPN1* allele under the control of the native PABPN1 promoter and a wild-type *PABPN1* allele showed that expression of proteins involved in the mitochondrial metabolism was significantly reduced in the transgenic mice compared to the wild-type mice, along with a significant reduction in the expression of SDHB (Complex II), COX1 (Complex IV), and ATP5A (Complex V) proteins in the affected muscles [66]. Our previous study revealed the abnormal accumulation of the expanded PABPN1 protein in the mitochondria is possibly associated with the mitochondrial abnormality in OPMD [67]. Thus, therapeutic approaches targeting the restoration of the mitochondrial functions should be considered.

## 6. Current Treatment and Prognosis

OPMD patients gradually develop a feeling of suffocation, regurgitation of food, and aspiration pneumonia as the disease progresses, although the rate of progression is different between the patients [68]. Repetitive aspiration may lead to death associated with malnutrition. OPMD reportedly has a little effect on the life expectancy because repeated aspiration occurs only at the more advanced stage. However, such conditions disturb the quality of life in patients [69,70]. A small proportion of the OPMD patients progressively get bound to wheelchairs, therefore, the clinicians should set up medical care services tailored to each patient.

Until now, no effective treatments have been developed for the disease. To control dysphagia, injecting botulin toxin into the cricopharyngeal muscle reportedly improved the swallowing function in 59% of the patients whereas 24% and 14% of the patients showed adverse dysphonia and deterioration of swallowing function, respectively [71]. Autologous myoblast transplantation accompanied by cricopharyngeal myotomy has been reported to enhance the quality of life in patients and preserve the swallowing function [65]. Moreover, since there are various mechanisms of dysphagia, a wide range of preventive approaches for aspiration should be employed. For ptosis, patients with a fair to good levator function usually undergo a levator-based procedure including levator advancement or resection, whereas those with a poor levator function undergo a frontalis suspension. However, the recurrence rate of ptosis in patients followed for at least 9 years has been reported to be 13% [72]. Thus, better approaches should be considered for long-term maintenance.

## 7. Patient Registry

Patient registry is an organized system that uses observational study methods to collect uniform data (clinical and other) to evaluate specified outcomes for a population defined by a particular disease, condition, or exposure, and that serves a predetermined scientific, clinical, or policy purpose(s) [73]. A flood of clinical trials is being planned and conducted for muscular dystrophies, including Duchenne and Becker muscular dystrophies. However, there are many challenges for clinical trials of such rare diseases, including OPMD. These include limited epidemiological data, the total number of patients, the natural history of the disease, assembling a certain number of patients, and adequate clinical outcome measures [74]. As solutions to these problems, patient registries are a helpful resource, particularly in cases of rare diseases.

Recently, the OPMD patient registry has been created at the University of New Mexico Health Sciences Center based on the rare disease registry standard created by the NIH Office of Rare Diseases Research [75]. It aims to understand how OPMD affects people’s lives; recruit participants for future studies on the disease; and establish communication among researchers, patients, and families (according to the website of the University of New Mexico). The achievements include an understanding of the relationship between physical and dysphagia symptoms and quality of life [69,70].

In Japan, the OPMD patient registry has not been established yet, although a national registry of Japanese DMD/BMD patients (REgistry of MUscular DYstrophy; Remudy. http://www.remudy.jp/ (accessed on 24 February 2021)) in collaboration with TREAT-NMD has been developed since 2009 [74]. By sharing the system environments, we are going to develop the unique patient registry of OPMD, which would enable us to collect epidemiological data, including the number of patients and the natural history of the disease. Registered fields will include the presence or absence of family history, consanguinity, muscle histology (rimmed vacuoles, intranuclear aggregates), (GCN)n repeat-length, height, weight, age at onset, initial symptoms, age at ptosis, diplopia, dysarthria, dysphagia, diet restrictions/alterations, tube feeding, lower proximal weakness, gait disorder, upper proximal weakness, any other complications including neuropathy, an examination of the respiratory (% vital capacity (VC) and % forced vital capacity (FVC)) and cardiac functions (% ejection fraction [EF] and % fractional shortening (FS)), electrocardiography, and serum CK levels (Table 1).

## 8. Conclusions

OPMD was first discovered in five French-Canadian patients in the second consecutive generation by Taylor in 1915. Since the identification of the causative gene in 1998, the research on the pathophysiology has progressed rapidly. However, clinicoepidemiological studies have not been performed in various parts of the world except endemic areas, including the Province of Quebec; Israel; and New Mexico. Based on the achievements of basic research, emergence of novel therapeutic approaches for OPMD is expected in the near future. Thus, it has become increasingly important to know patients’ unmet medical needs, realize therapeutically targetable symptoms, and identify indices of therapeutic efficacy. Our attempt to establish a unique patient registry of OPMD would be a helpful tool to address these urgent issues.

## Figures and Tables

**Figure 1 jcm-10-01375-f001:**
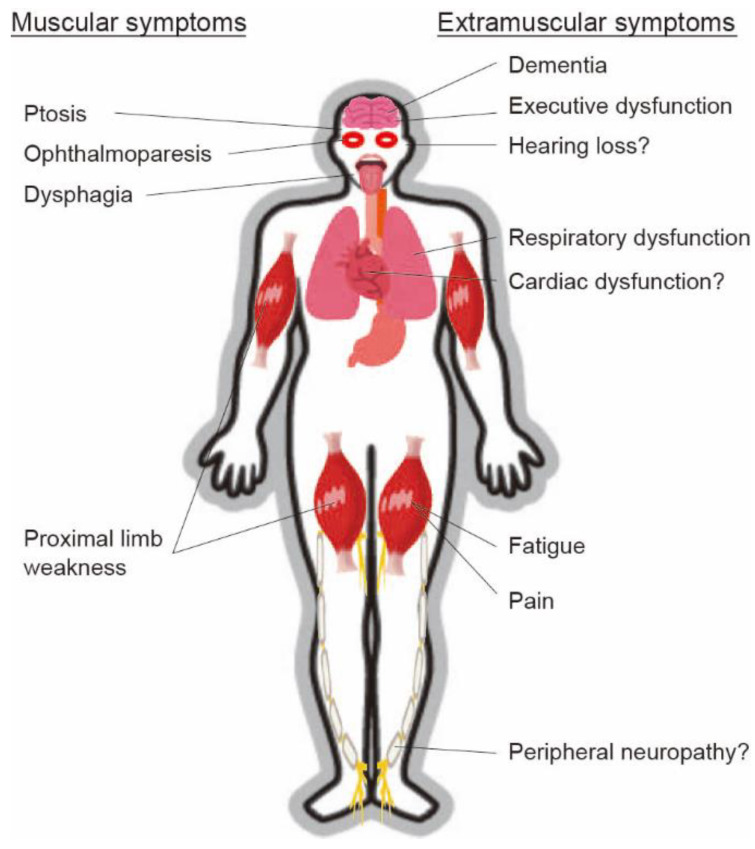
Various muscular and extramuscular symptoms of oculopharyngeal muscular dystrophy (OPMD). Patients with OPMD can manifest various muscular and extramuscular symptoms.

**Figure 2 jcm-10-01375-f002:**
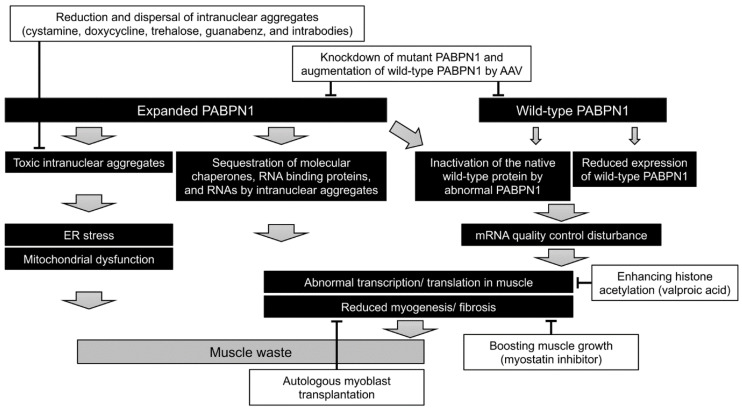
Pathogenesis and therapeutic approach for OPMD. Several therapeutic strategies have been proposed based on the pathogenesis of OPMD, such as (1) knockdown of mutant PABPN1 and/or augmentation of wild-type PABPN1 using adeno-associated virus (AAV) vectors; (2) boosting muscle growth using myostatin inhibitor; (3) reduction and dispersal of intranuclear aggregates using cystamine, doxycycline, trehalose, guanabenz, and intrabodies; and (4) autologous myoblast transplantation combined with cricopharyngeal myotomy.

**Table 1 jcm-10-01375-t001:** Registered fields for the proposed nation-wide OPMD patient registry in Japan.

Basic Items	Name
	Date of birth
	Nationality
	Address
	Participation in patients’ association
	Proposal for clinical trials
Clinical items	Presence or absence of family history
	Consanguinity
	Muscle histology (rimmed vacuoles, intranuclear aggregates)
	(GCN)n repeat-length
	Height and weight
	Age at onset
	Initial symptoms
	Age at ptosis
	Age at diplopia
	Age at dysarthria
	Age at dysphagia
	Age at diet restrictions/alterations
	Age at tube feeding
	Age at lower proximal weakness
	Age at gait disorder
	Age at upper proximal weakness
	Any other complications including neuropathy
	Respiratory (% vital capacity and % forced vital capacity)
	Cardiac functions (% ejection fraction and % fractional shortening)
	Electrocardiography
	Serum CK levels

## Data Availability

Not applicable.

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
