# Peer review of "Recent Progress in Oculopharyngeal Muscular Dystrophy"

_jcm, 2021, doi:10.3390/jcm10071375_

Round 1

Reviewer 1 Report

An interesting review on OPMD, enriched by the addition of the presentation of a disease registry project.

I have a few minor comments:

  • Lines 46-48: Can the author clarify this sentence? Are only 6 patients with OPMD diagnosed at the Author's Institute? Are the 29 patients affected by sIBM to be correlated with the differential diagnosis?
  • Lines 80: the author refers to the serum level of CK as a possible biomarker of disease; can the author cite an article published about it? Does the author have any personal data not reported in this review?
  • PABPN1 when referring to the gene should be italics

Finally, a suggestion for the disease registry project: it would be interesting to collect systematic data on the presence of neuropathy in patients with OPMD.

Author Response

  1. Lines 46-48: Can the author clarify this sentence? Are only 6 patients with OPMD diagnosed at the Author's Institute? Are the 29 patients affected by sIBM to be correlated with the differential diagnosis?

 I would like to thank the reviewer for the helpful comment. To clarify this sentence, I rewrote it and removed the description regarding sIBM as follows.

Lines 46-48: “In our single-institution experience, only 6 patients with OPMD were diagnosed during the last decade for which 414 patients underwent muscle biopsy.” was corrected.

  1. Lines 80: the author refers to the serum level of CK as a possible biomarker of disease; can the author cite an article published about it? Does the author have any personal data not reported in this review?

I would like to thank the reviewer for the careful comment. I cited the reference by Richard et al. [29], which mentioned that CK levels were normal in patients with a repeat number <14, while this level increased with the number of repeats. I included that the observation that levels of myogenic enzymes gradually decrease and are normalized as the disease advances was based on unpublished personal data.

Lines 80-84: “Serum levels of creatine kinase (CK) are slightly elevated in mild cases, but the levels are highly increased in more severe cases [29]. However, the levels of myogenic enzymes gradually decrease and are normalized as the disease advances (unpublished personal data), suggesting the limitation of using the enzymes as biomarkers for evaluating the disease progression and therapeutic efficacy.” was corrected.

  1. PABPN1 when referring to the gene should be italics.

 According to the reviewer’s suggestion, I rewrote PABPN1 in italics.

  1. Finally, a suggestion for the disease registry project: it would be interesting to collect systematic data on the presence of neuropathy in patients with OPMD.

We truly appreciate the reviewer’s comment. I agree to collect systematic data on the presence of neuropathy in patients with OPMD. In the text and table 1, I added any other complications including neuropathy.

Lines 253-262: “Registered fields will include the presence or absence of family history, consanguinity, muscle histology (rimmed vacuoles, intranuclear aggregates), (GCN)n repeat-length, height, weight, age at onset, initial symptoms, age at ptosis, diplopia, dysarthria, dysphagia, diet restrictions/alterations, tube feeding, lower proximal weakness, gait disorder, upper proximal weakness, any other complications including neuropathy, and examination of the respiratory (% vital capacity [VC] and % forced vital capacity [FVC]) and cardiac functions (% ejection fraction [EF] and % fractional shortening [FS]), electrocardiography, and serum CK levels (Table 1).” was corrected.

Reviewer 2 Report

This author provides a very comprehensive review of oculopharyngeal muscular dystrophy (OPMD). The authors present a novel idea for setting up a nation-wide OPMD disease registry in Japan to better identify potential patients that might benefit from therapeutic approaches to treat OPMD. The epidemiology, presentation, diagnosis, pathogenesis and current treatments are reviewed, as are newer therapeutic approaches, with autologous myoblast transplantation as a primary treatment to improve swallowing. The authors describe this new patient registry and the specific patient information to be included to better understand the natural history of the disease and help determine potential treatment efficacy. The writing is clear. A major strength of this paper is the detailed literature review, which is extensive and current, particularly as relates to the newer therapeutic strategies. In addition, the specific fields for the patient registry are well thought-out.  Table and figures are clear and appropriate. This is a nice addition to the literature concerning OPMD. In addition, the prospect of a patient registry in Japan will potentially improve the understanding of the disease progression, the natural course as it relates to swallow impairment, and the utility of newer treatment strategies on disease course and functioning. I only had a few minor comments:

  1. p 122, lines 117-119. the authors state: “…fiber-optic endoscopic evaluation identified a significant impairment in salivary pooling and pharyngeal clearance”. Pharyngeal clearance is a physiologic component of the swallow and can be impaired. However, salivary pooling is the impairment itself in the swallow, so wording should be modified. Perhaps re-word to say something like “…significant impairment in pharyngeal clearance resulting in salivary pooling..”.
  2. Perhaps the authors could include “Age at dysphagia” to include first symptoms, or diet restrictions/alterations, as well as “Age at dysphagia (tube feeding). Individuals often go for a period of time with dysphagia requiring diet alteration prior to needing tube feeding.

Author Response

  1. p 122, lines 117-119. the authors state: “…fiber-optic endoscopic evaluation identified a significant impairment in salivary pooling and pharyngeal clearance”. Pharyngeal clearance is a physiologic component of the swallow and can be impaired. However, salivary pooling is the impairment itself in the swallow, so wording should be modified. Perhaps re-word to say something like “…significant impairment in pharyngeal clearance resulting in salivary pooling.”.

As suggested by the reviewer, I rewrote the sentence as follows.

Lines 121-123: “In our investigation, the fiber-optic endoscopic evaluation identified a significant im-pairment in pharyngeal clearance resulting in salivary pooling (data submitted).” was corrected.

  1. Perhaps the authors could include “Age at dysphagia” to include first symptoms, or diet restrictions/alterations, as well as “Age at dysphagia (tube feeding). Individuals often go for a period of time with dysphagia requiring diet alteration prior to needing tube feeding.

I would like to thank the reviewer for the careful comment. I separated the data collection into three parts: ages at dysphagia, diet restrictions/alterations, and tube feeding.

Lines 253-262: “Registered fields will include the presence or absence of family history, consanguinity, muscle histology (rimmed vacuoles, intranuclear aggregates), (GCN)n repeat-length, height, weight, age at onset, initial symptoms, age at ptosis, diplopia, dysarthria, dysphagia, diet restrictions/alterations, tube feeding, lower proximal weakness, gait disorder, upper proximal weakness, any other complications including neuropathy, and examination of the respiratory (% vital capacity [VC] and % forced vital capacity [FVC]) and cardiac functions (% ejection fraction [EF] and % fractional shortening [FS]), electrocardiography, and serum CK levels (Table 1).” was corrected.